# Reproductive Anatomy of Chondrichthyans: Notes on Specimen Handling and Sperm Extraction. II. Sharks and Chimaeras

**DOI:** 10.3390/ani11082191

**Published:** 2021-07-23

**Authors:** Pablo García-Salinas, Victor Gallego, Juan F. Asturiano

**Affiliations:** 1Grupo de Acuicultura y Biodiversidad, Instituto de Ciencia y Tecnología Animal, Universitat Politècnica de València, 46022 València, Spain; pau.salinas82@gmail.com (P.G.-S.); vicgalal@upvnet.upv.es (V.G.); 2Associació LAMNA per a L’estudi dels Elasmobranquis a la Comunitat Valenciana, Fraules 10, 13, 46020 València, Spain

**Keywords:** fish reproduction, aquarium, ex situ conservation, reproductive assisted techniques, artificial insemination

## Abstract

**Simple Summary:**

Sperm extraction and artificial insemination may serve ex situ conservation initiatives for threatened sharks and related species. A comparison of the reproductive anatomy of eight chondrichthyans is presented in this study, emphasizing the important differences when performing these reproductive techniques. Additionally, we show how to obtain sperm samples from both living and dead specimens using cannulation, abdominal massage, or oviducal gland extraction. These tools can improve the success of breeding programs developed in aquaria and research facilities.

**Abstract:**

The chondrichthyan fishes, which comprise sharks, rays, and chimaeras, are one of the most threatened groups of vertebrates on the planet. Given this situation, an additional strategy for the protection of these species could be the ex situ conservation projects developed in public aquaria and research centers. Nevertheless, to increase sustainability and to develop properly in situ reintroduction strategies, captive breeding techniques, such as sperm extraction and artificial insemination, should be developed. These techniques are commonly used in other threatened species and could be also used in chondrichthyans. However, the different reproductive morphologies found in this group can complicate both processes. Therefore, a comparison of the reproductive anatomy of eight distinct chondrichthyans, with an emphasis on those important differences when performing sperm extraction or artificial insemination, is carried out herein. Sharks and chimaeras belonging to the Scyliorhinidae, Carcharhinidae, Centrophoridae, Etmopteridae, Hexanchidae, and Chimaeridae families were obtained from commercial fisheries, public aquaria, and stranding events. In addition, the process of obtaining viable sperm samples through cannulation, abdominal massage, and oviducal gland extraction is described in detail for both living and dead animals.

## 1. Introduction

The Chondrichthyes are a group of vertebrates that appeared more than 400 million years ago. Nowadays, this group is an ecologically diverse group with great importance in the regulation of the ecosystems where these animals inhabit [1,2]. The class Chondrichthyes comprises 1472 species classically divided into the holocephalans (subclass Holocephali), which are commonly named chimaeras, and the elasmobranchs (subclass Elasmobranchii), commonly named sharks and rays [3]. The chimaeras are the smallest of these three divisions in terms of the number of extant species, and currently there are 57 species described [3]. All chimaeras are marine animals, and very few species inhabit shallow waters, most species relegated to the deep waters (>200 m) despite their global distribution in the past [4]. Chimaeras have significantly different features compared to the elasmobranchs, such as the fusion of the lower jaw to the cranium (hence their name Holocephalans, “complete heads”) and non-replaceable tooth plates as teeth [4]. In general, holocephalans are a less studied group than their relatives, the rays and sharks. This last group, the sharks, is perhaps the most recognizable group among Chondrichthyans despite not being as numerous in terms of the number of species as the group formed by the rays and skates [5,6].

Regarding their conservation status, Chondrichthyes possess life histories that make them sensitive to elevated anthropic pressure, threatening their populations [7,8]. In fact, chondrichthyan extinction risk is higher than for most other vertebrates, and only one-third of the species assessed are considered safe according to IUCN red list criteria [9]. The situation is particularly sensitive in places such as the Mediterranean Sea, a key hotspot of extinction risk, where half the species of rays and sharks face an elevated risk of extinction [10]. Among the drivers for the global decline of its populations, overfishing (intentional or incidental) and habitat destruction are the main causes [9,10,11].

As mentioned above, to understand the current global situation of chondrichthyan populations, their reproductive strategies and life histories should be noted. Sharks and their relatives show larger body sizes, slower sexual maturity, longer gestation periods, higher maternal investment, and fewer offspring than other fishes [11,12]. Chondrichthyan species reproduction modes are diverse and can be divided according to the nutrition of the embryos. Lecithotrophic modes include oviparity (such as the catsharks of the family Scyliorhinidae, or the entire subclass Holocephali) and yolk sac viviparity (such as Hexanchiformes), where the only nourishment comes from their yolk sack. Matrotrophic modes include an additional nourishment source at some point of the embryo development, in the form of lipid histotrophy, unfertilized eggs (or fertilized eggs in the extreme case of the sand tiger shark *Carcharias taurus*), or the formation of a placenta (such as some Carcharhiniformes) [13,14,15].

In aquaculture industries, the reproductive factors and complex life histories mentioned above have discouraged captive breeding programs [13], but not in aquaria facilities, either public or private. The reproduction in captivity of elasmobranchs and chimaeras has been reported for some species [16,17,18], but these events have traditionally relied more on natural mating rather than on the use of reproductive techniques [19]. A potentially useful technique in breeding programs is the artificial insemination of females, but to ensure its success, a reliable supply of sperm is required, especially in the case of endangered species [19,20,21,22,23,24,25]. Although the obtention of sperm has been previously achieved for several shark and chimaera species [19,24,25,26,27,28,29], the procedures of extraction may vary between the different groups. In live animals, the most common procedures for sperm obtention have been cannulation and abdominal massage [19,22,29], but these techniques should consider the morphology and location of the reproductive structures, such as the seminal vesicles and urogenital papillae, to be truly effective.

Due to their position as one of the oldest groups of vertebrates [30], Chondrichthyes have been previously used as animal models for physiological and morphological studies [31,32,33,34]. Moreover, certain aspects of their reproductive morphology, such as the form and function of their intromittent organs [35,36,37,38] or gonads [39,40,41,42,43,44,45,46], have received attention from researchers and are well studied. However, some details about the morphology of certain reproductive structures, which are important during the use of reproductive techniques, have not been previously considered for sharks and chimaeras. Thus, the aim of this study is to offer an anatomical guide intended to be useful during sperm obtention procedures, as well as propose preliminary indications to be considered during artificial insemination. This practical guide complements previous work focused on the anatomy of batoids (rays, skates, and close species) and the use of reproductive techniques on them [47]. The tools presented in both studies are intended to delve into the development of reproductive techniques, such as sperm cryopreservation [29] or artificial insemination [25], some of the important steps toward successful and sustainable breeding programs.

## 2. Materials and Methods

### 2.1. Origins of the Specimens

Males and females of eight Chondrichthyan species belonging to the orders Carcharhiniformes (*n* = 35), Hexanchiformes (*n* = 1), Squaliformes (*n* = 5), and Chimaeriformes (*n* = 4) were studied (Table 1). Some of the species, such as the small-spotted catshark (*Scyliorhinus canicula*) and blackmouth catshark (*Galeus melastomus*), were available in fish markets and from commercial fishing vessels’ by-catch. The rabbitfish (*Chimaera monstrosa*), little gulper shark (*Centrophorus uyato*), and velvet belly lanternshark (*Etmopterus spinax*) specimens were obtained from the bottom trawling by-catch. The blue sharks (*Prionace glauca*) and the bluntnose sixgill shark (*Hexanchus griseus*) specimens appeared stranded on beaches and were recovered by the Comunitat Valenciana Stranding Network (Valencian Community, Valencia, Spain).

Other animals were held alive in captivity as part of the zoological collection of a public aquarium (Oceanogràfic, València). Both nursehound (*Scyliorhinus stellaris*) and small-spotted catsharks (*Scyliorhinus canicula*) were kept separately in two 8000 L tanks with recirculating seawater (temperature: 16–18 °C; salinity: 35–37‰), and fed twice a day with herring, squid, and shrimps. The handling of the specimens was authorized by the Ethic Committee of Fundación Oceanogràfic (project reference: OCE-16-7777719), and the sperm extraction process was carried out under the supervision of their veterinary team. The maturity of all the specimens was determined by their gonad development, grade of their clasper calcification and development of secondary copulatory organs (in *C. monstrosa*), and their size, according to the literature [48].

### 2.2. Dissection Procedure

On dead specimens, a dissection was performed to specifically gain access to the reproductive system (Figure 1). The procedure focused only on the reproductive structures, following and adapting the dissection procedures and terminology used by other authors [49,50,51]. In the case of medium-size and small sharks, the animals were flipped dorsally, exposing their ventral surface. A small incision was made over the coracoid bar of the pectoral girdle and a longitudinal cut was made along the ventral midline toward the pelvic girdle, over the cloaca (Figure 1A). If needed, a cut made transversely to the midline, over the pectoral girdle, and another cut over the pelvic girdle, allowed us to fully expose the pleuroperitoneal cavity (Figure 1B).

In large sharks and chimaeras, where it was not possible to fully expose the ventral part of the animal due to its size or the morphology of its dorsal fins, the process was slightly different. First, a small incision with scissors was made at one side of the ventral midline, near the pectoral fin base, over the coracoid bar. Through this incision and with the help of a scalpel blade, a curved cut was made toward the pelvic girdle. The cut followed a slight curve from the ventral midline to the lateral and then back again to the ventral midline, to create a kind of flap. Note that to avoid damage to the inner organs, forceps were used to keep the abdominal wall (skin, muscle, parietal peritoneum) elevated. The pleuroperitoneal cavity was exposed by pulling out the flap to one side.

To access the reproductive system, the hepatic lobes were drawn forward and removed by cutting through their cranial attachments (hepatic ducts and falciform ligament) (Figure 1C), trying to avoid damage to this organ and to avoid leakage of the oil and bile present in the liver (Figure 1D). A cut close to the rectal gland was made on the rectum to separate the intestine (spiral valve), and close to the pericardiac cavity, through the esophagus, to separate the stomach (Figure 1E). Then, the mesogaster and caudal mesenteries were cut to completely remove the internal parts of the digestive tract. Special care was taken to not damage the mesorchium holding the ovaries and the testes.

In sharks, a cut through the cloaca (Figure 1F) and puboischiadic bar was made to expose the caudal part of the urogenital system (seminal vesicle, urogenital sinus, urogenital papilla in males, or uterine sphincters and urinary papilla in females). In rabbitfish, both urogenital papilla and uterine sphincters are external, so there was no need for this procedure. Moreover, in some species the removal of cloacal lips to access the urogenital/urinary papilla was necessary.

### 2.3. Description of Reproductive Structures

Detailed photographs were taken with a macro lens camera throughout the dissection procedure of every species and illustrated notes were taken. The focus of each dissection was on (i) determining how to gain easy access to the urogenital papilla, (ii) observing the number and disposition of urogenital pores, (iii) observing the urogenital sinus morphology, and (iv) accessing the seminal vesicle/uterus. Plastic tubes with different gauges (0.5–2 mm in diameter) were used as probes during the dissection to confirm the access and connections from the external part of the reproductive system (urogenital/urinary pores and papilla) to the internal part (urogenital sinus, seminal vesicle, and uterus). The information obtained was combined and used to propose a general morphological scheme of an ideal male and female shark.

### 2.4. Sperm Collection

#### 2.4.1. *In Vivo* Sperm Extraction

Before sperm extraction, tonic immobility was induced in a small-spotted catshark (*S. canicula*) and a nursehound (*S. stellaris*) to minimize struggling and reduce stress during handling [52,53,54]. The animals were held in an upside-down position (with their ventral region exposed) with their mouth and gill slits submerged, while gentle pressure was exerted on their snouts. Then, with the cloaca emerged, gentle pressure on the abdominal area (over the location of the seminal vesicle on the sides of the animal) was exerted to make sperm flow through the urogenital papilla. Then, sperm was immediately collected using a sterile syringe or pipette and transferred to sterile tubes after collection.

#### 2.4.2. *Postmortem* Sperm Extraction

The cloacal area in sharks, and the posterior portion of the body in chimaeras, was cleaned of mucus and other biological remains (such as blood, mud, and the remains of other organisms, which may be found in animals obtained from fisheries) using marine water. Three different methods were used to obtain sperm from dead males in every species: (i) abdominal massage on the ventral region immediately anterior to the pelvic girdle, or by pressing around the urogenital papilla in the cloacal cavity with the fingers or with curved pincers (only in sharks); (ii) accessing by dissection and stripping directly on the seminal vesicle. In both cases, the sperm flowing from the urogenital papilla was immediately collected using a sterile syringe, plastic tube, or a pipette; and (iii) through cannulation, by introducing a polyurethane cat catheter (BUSTER cat catheter, 1.0 × 130 mm, Kruuse, Langeskov, Denmark) or a PVC nasogastric tube (Feeding Probe L/RX CH-05 2.67 × 50 mm, JMEDIS, Cádiz, Spain) through the appropriate pore on the urogenital papilla. A sterile lubricating jelly with antiseptic (Optilube ActiveTM, Optimum Medical, Leeds, UK) was used to facilitate the insertion of the tube. Tubes and catheters were continuously rotated inside the seminal vesicle to avoid the clogging of their orifices.

The nidamental glands from small-spotted catshark (*S. canicula*), nursehound (*S. stellaris*), blue shark (*P. glauca*), and rabbitfish (*C. monstrosa*) females were obtained (Figure 2A). The organs were carefully removed during the dissection by cutting through the anterior and posterior oviduct. Then the glands were externally cleaned with artificial seawater to remove the blood and biological remains (Figure 2B). Care was taken during the cleansing process to avoid the passing of seawater into the lumen of the glands. The organs were split along its cranial–caudal axis to expose its lumen (Figure 2C). A gentle scrape was done over its luminal epithelium using a scalpel blade (Figure 2D). Then, a pearly mucus was collected over the edge of the blade. The mucus was diluted in an artificial seminal plasma extender described by García-Salinas et al. (2021) [29]. To summarize, the main components of the extender (in mM; 433 urea, 376 NaCl, 120 trimethylamine N-oxide (TMAO), 8.4 KCl, 50 glucose, 7 CaCl_2_.2H_2_O, 3.5 NaHCO_3_, 0.08 Na_2_SO_4_, 1.4 MgSO_4_) were kept in balance with physiological fluids by adjusting pH to 6.5 and the osmolality to 1000 mOsm/kg.

## 3. Results and Discussion

### 3.1. Female General Anatomy: Sharks

It should be noticed that there is a wide variety of anatomical differences between the species studied, but this study aims to highlight only those that may be important while using specific techniques, such as sperm extraction and artificial insemination. Thus, other anatomical differences have not been considered. However, even though the abdominal pores are not part of the reproductive system, its misidentification may cause errors during cannulation, so its description is offered.

Externally, shark females possess a cloaca located between the pelvic fins, where the urinary, reproductive, and digestive system converge (Figure 3). Next to the caudal margin of the cloaca, the abdominal (or celomic) pores are found. These small orifices closed by a sphincter connect the pleuroperitoneal cavity with the exterior and may allow the removal of fluid from the inner cavity [51]. In females, the urinary system ends in the urinary papilla (the term urogenital papilla should be limited to males). Inside the cloaca, the caudal end of the uteri can be independent or converge and fuse in a common cervix (or urogenital sinus). The cranial orifice inside the cloaca is the caudal section of the digestive system: the rectum and the anus.

Internally, females possess a pair of reproductive tracts distributed along with the entire pleuroperitoneal space. The ovaries are embedded in the epigonal gland and located on the cranial part of the pleuroperitoneal cavity. Depending on the species, the ovaries can be paired, fused in a single ovary or vestigial, with only one ovary developed [13,15,38]. The oogenesis is produced in the ovaries, and the mature oocytes are released into the celoma and transported to the ostium by the action of cilia. The ostium is a funnel-like structure to allow the passage of the unfertilized oocytes into the anterior part of the oviduct and, then, to the oviducal gland (the terms shell gland and nidamental gland are commonly used, although in sharks, eggshells or embryo envelopes are not always produced). That gland is responsible for the storage of the sperm, the oocyte fertilization, and the formation of the egg case [55]. Thus, in oviparous species, the shell gland is proportionally bigger than in ovoviviparous or viviparous species. The gland also supposes the terminological division of the oviduct in the anterior (pre-oviducal gland) and posterior (post-oviducal gland) oviduct. In some species, a sphincter-like structure, the isthmus, can seal the oviducal gland from the uterus. Internally, females have one or two functional uteri depending on the species [13,15,38], where the embryos develop, or the eggs are kept until they are laid.

### 3.2. Female Anatomy: Chimaera monstrosa

Externally, *C. monstrosa* does not possess a common chamber (cloaca) where the digestive, urinary, and reproductive systems converge, but a single opening for the caudal portion of the digestive system, located between the pelvic fins (Figure 4) [46]. Two small pores closed by a sphincter sit along with this orifice, the abdominal (or celomic) pores that allowed an exchange of fluids between the exterior and the pleuroperitoneal cavity [51]. The entrances to the uteri are two independent orifices closed by a sphincter located at the base of the pectoral fins. In juvenile females, the orifices are narrow and hard to find, while in mature females, they are easier to locate.

Internally, their reproductive system is composed of paired structures arranged longitudinally, in the same way as in the general model for sharks and rays. Following the longitudinal axis, two ovaries, a single ostium, two oviducts (divided into anterior and posterior) with an oviducal gland each, two isthmus, two functional uteri ended by the cervix, and a sphincter in the posterior section can be found. As in the rest of oviparous chondrichthyans [56], the oviducal gland of *C. monstrosa* is well developed, as at present, all chimaera species lay hard egg cases, despite their other past reproductive strategies [57].

### 3.3. Female Comparative Anatomy

In sharks, there are no great differences between the species studied (Figure 5) regarding the general morphological structure of the female reproductive system. The overall morphology of the reproductive system in *Scyliorhinus canicula*, *S. stellaris*, and *Galeus melastomus* (Figure 5A) is similar, and closely resembles that of the ideal model proposed for sharks (Figure 3). However, in the two *Scyliorhinus* species, the abdominal pores lay over the metapterygium of the pelvic fins instead of inside the cloaca, while in blackmouth catshark (*G. melastomus*) the pores are located inside the cloaca. All three species belong to the family Scyliorhinidae, one of the six shark families where oviparous sharks have been described [14]. As happens with oviparous sharks, their oviducal glands are well developed, especially in *G. melastomus* where multiple (or retained) oviparity occurs [58].

In the blue shark (*P. glauca*), three papillae are located in the cloaca (Figure 5B). The central one is the urinary papilla and it connects with the urinary system. The two others are located on both sides of the cloaca and contain the abdominal pores (or abdominal papillae in this case). Unlike the rest of the species observed, only the right ovary is developed and functional in blue sharks. The oocytes pass into the heart-shaped oviducal gland, where they are fertilized and encapsulated in a thin, soft capsule before passing into the uterus. *P. glauca* is a species where placental viviparity has been described; thus, the oviducal gland and the oocytes are proportionally smaller than in other species [59]. Lastly, the two entrances to the uterus fuse into a large cervix, where the pups pass momentarily before birth.

In the velvet belly lanternshark (*Etmopterus spinax*), two functional ovaries and uteri can be found. The uteri converge in a small common cervix located in the cranial part of the cloaca, next to the urinary papilla. Two small abdominal pores are located at the end of the cloaca, covered by the pelvic fins. Unlike the previous species, *E. spinax* is an ovoviviparous species (also called viviparous lecithotrophic or aplacental viviparous) [60] and does not produce any case covering the fertilized ova; thus, their oviducal gland is reduced and scarcely developed with no lateral expansions [61].

The rabbitfish (*Chimera monstrosa*) shares the same anatomic features as the oviparous sharks examined, such as well-developed paired ovaries and relatively big oviducal glands. However, instead of uteri converging inside the cloaca, the uterine openings of *C. monstrosa* are located behind the pectoral fins. The absence of cloaca is described for most Holocephali [46,62,63], but some species such as the elephant fish (*Callorhynchus milli*) [64], the cockfish (*Callorhynchus callorhynchus*) [65], and the Eastern Pacific black ghost shark (*Hydrolagus melanophasma*) [37] possess a common cloaca where the anus and the uterine openings are found. Moreover, a sperm pouch for sperm storage can be found on these species, but it is absent in *C. monstrosa*. Another reproductive trait, the prepelvic clasper pouch (present in female *Callorhynchus* species) [63,64,65] is not present in *C. monstrosa* females.

### 3.4. Anatomic Notes for Artificial Insemination

Currently, artificial insemination is not a widespread technique in sharks, and there is not any report about the use of this technique in chimaeras. To date, artificial insemination for shark reproduction has been used in the brown banded bamboo sharks (*Chiloscyllium punctatum*), the zebra shark (*Stegostoma fasciatum*), the cloudy catshark (*Scyliorhinus torazame*), and the white-spotted bamboo shark (*Chiloscyllium plagiosum* [22,23,24,25]), although the technique has also tried in other species such the sand tiger shark (*Carcharias taurus*) and the broadnose sevengill shark (*Notorynchus cepedianus* [19]).

The process involves depositing viable sperm into the female reproductive tract by inserting a catheter through the uterine sphincter into the cervix or into the oviducal gland or the posterior oviduct (intrauterine). However, it has been proposed that intrauterine or oviducal insemination results in higher fertilization rates than cervix insemination [19,22]. The catheter with the sperm sample should be inserted through different anatomical structures (such as cloacal morphology, cervix, or uterine sphincters) that can hamper the overall insemination procedure [21]. Thus, to perform artificial insemination protocols, the female reproductive system must be well known, and in the same way, the morphology of the male reproductive system when extracting sperm.

### 3.5. Male General Anatomy: Sharks

Externally, male sharks have paired prolongations of the pelvic fins called claspers (or myxopterygium) used as intromittent organs for internal fertilization [13,34,35,36,37] (Figure 6). Claspers are located at the inner margin of the pelvic fins, forming a tube-like structure with a ventral groove known as hypopyle. The sperm flows through this groove from the male urogenital papilla to the female cloaca, and then into the uterine sphincter. To impulse the sperm, most sharks possess paired muscular structures at the base of the claspers, under the ventral skin, called siphon sacs [37,66]. The siphon sacs secrete serotonin (5-hydroxytryptamine), capable of producing muscle contractions in the uterus of females, which could favor the intrauterine transport of sperm [66]. Because multiple mating in sharks is well known, it has also been proposed that siphon sacs may play a role in sperm competition by washing rival sperm from the cloaca of females [67,68].

The cloaca is located between the pelvic fins. Typically, it is covered by the inner margins of the pelvic fins (the metapterygia), or in some species by two cloacal lips. Inside the cloaca are located the anus (caudal portion of the digestive system) and the urogenital papilla. Depending on the species, on the tip of the urogenital papilla, there are one or two pores through which the sperm and urine flow. Lastly, two abdominal pores connecting the pleuroperitoneal cavity with the exterior are located inside or near the cloaca, closed by a sphincter [51].

Internally, the male reproductive system is located along the pleuroperitoneal cavity. The paired testes, embedded in the epigonal organ, are located at the cranial end of the reproductive tract. The testes are connected via efferent ductules (or ductuli efferentes) with two genital tracts composed of the cranial convoluted part called epididymis, the ductus deferens (also called the Wolffian duct or vas deferens) [13], and the caudal part called seminal vesicle (or ampulla) where the sperm is stored. The Leydig gland is adjacent to the ductus deferens and empties its content on it and on the epididymis. The gland is formed by the cranial part of the mesonephros and produces the seminal fluid and the matrix where the spermatozeugmata, or the spermatophores, are formed [14,37,38,69]. In some species, a pouch-like structure can be found on the ventral surface of the seminal vesicle, the sperm sac, where mature sperm is stored until mating [41].

### 3.6. Male Anatomy: Chimaera monstrosa

The internal male reproductive system in *C. monstrosa* (Figure 7) is similar to that of sharks in many aspects [63]. Two paired testes located in the cranial region of the pleuroperitoneal cavity are connected with two genital tracts (which includes the epididymis, the ductus deferens with the Leydig gland and the seminal vesicle) by efferent ductules (or ductuli efferentes). The two seminal vesicles (or green glands [65]) are divided by an isthmus into a cranial section, first whitish and opaque and then greenish and translucent, and a grayish posterior section. The greenish coloration of the middle part of the seminal vesicle is a trait present in Chimaeriformes [46,65,70]. These caudal sections converge in a common sinus before forming the urogenital papilla in the exterior of the body.

The external reproductive traits are multiple in *C. monstrosa*. First, the claspers are located at the base of the pelvic fins and receive the sperm from the external urogenital papilla. The claspers are three-lobed, with dermal denticles along its terminal region. Moreover, in *C. monstrosa* males, there are two slits located in the cranial part of the pelvic fins. In juvenile males, the slits are small and almost closed, but in adult males, the slits house prepelvic claspers, two serrated blade-like structures that emerge to grab the female during mating [18,46,54]. Another reproductive external trait is the cranial tenaculum. The tenaculum is a single mallet-like structure in the forehead of male chimaeras, with dermal denticles in its extreme. As in the case of prepelvic claspers, the tenaculum is used during reproduction to grab the females [18].

### 3.7. Male Comparative Anatomy

The overall morphology of the structures which can be relevant during sperm extraction is well preserved in all the species studied (Figure 8). However, there are some significant differences. Members of the family Scyliorhinidae studied (*S. canicula*, *S. stellaris*, *G melastomus*) are quite similar (Figure 8A). The three species have two independent seminal vesicles converging in a single sinus and urogenital papilla, but both *Scyliorhinus* species possess abdominal pores near the inner margin of the pelvic fins, while the abdominal pores in *G. melastomus* are slightly more centered near the cloaca. The siphon sacs in the three species are also visible under the skin cranial to the pelvic fins. In *P. glauca* the siphon sacs are also easily visible and are clearly associated with each clasper, but there are several morphological differences with the previous group. In this species (Figure 8B) the abdominal pore takes the form of a papilla (abdominal papilla) located outside the cloaca. Inside, the two caudal parts of the deferent ducts converge in a wide common urogenital sinus along the ureters of the urinary system. Both the urogenital sinus and the deferens ducts are capable of storing a large amount of sperm. The urogenital sinus opens to the exterior through the urogenital papilla located in the middle of the cloaca. Even with these differences, the species are close relatives and belong to the same order Carcharhiniformes.

In the case of the little gulper shark (*Centrophorus uyato*), a deep-water shark that belongs to the order Squaliformes, the internal reproductive structures (Figure 8C) closely resemble those of the previous models observed: two paired testes, reproductive ducts, and independent seminal vesicles converging in a urogenital sinus. The external reproductive traits, however, showed several differences. The cloaca is partially covered by two cloacal lips, and only the tip of the urogenital papilla is visible if the lips are not separated. Claspers are proportionally shorter and thinner, with internal spines that unfold inside the female tract during mating. Moreover, siphon sacs are not visible in the ventral region, as in the previous species, but there are two folds under the ventral surface of the pelvic fins (also called siphons) with the same function. The absence of ventral siphon sacs also occurs in the bluntnose sixgill shark (*Hexanchus griseus*), another deep-water species belonging to the order Hexanchiformes (Figure 8D). In this species, a sac-like structure along the clasper (called clasper sac) can be found. This structure, a unique feature of Hexanchiformes, can inflate and function like the siphon sac of other elasmobranchs. Moreover, in this species the claspers lie in the inner rear margin of the pelvic fins which form a scroll, absent in females [71]. As in *C. uyato*, the abdominal pores in *H. griseus* are located at the base of the cloaca, but in this species as two abdominal papillae. Unlike the rest of the species studied, the ductus deferens and seminal vesicles do not converge in a common urogenital sinus. Instead, the two reproductive tracts are independent even in the urogenital papilla, where two different pores are located, one for each tract. This difference has also been described in the broadnose sevengill shark (*Notorynchus cepedianus*) [72], another member of the Hexanchiformes, suggesting that it could be a common trait for this order.

*C. monstrosa* is the only holocephalan studied, but with some exceptions, the overall internal morphology of the reproductive system is shared with the rest of the species observed (Figure 8E). The epigonal organ, present in elasmobranchs, is absent or cannot be easily identified in *C. monstrosa* and probably in the rest of holocephalans [46,65,69,73]. The division of the seminal vesicle into two sections separated by an isthmus does not occur in elasmobranchs but appears in other holocephalans such as the spotted ratfish (*Hydrolagus colliei* [73]) and *H. melanophasma* [65], though neither in *C. callorhynchus* [64] nor *Chimaera phantasma* [70]. The greatest difference between *C. monstrosa* and the rest of species studied appears when observing their external reproductive traits such as the clasper morphology, the absence of cloaca, and the presence of prepelvic claspers and tenaculum.

### 3.8. Anatomic Notes for Sperm Extraction

In mature sharks, the abdominal massage is easily performed by pressing in the pelvic region and the lateral of the body, especially in small and medium-size animals. The idea behind this procedure is to be able to put pressure on the seminal vesicles or the area around them. The technique is useful in mature males that have abundant sperm stored in their seminal vesicles. In animals outside of the peak of the reproductive season, or in immature males, this technique is less useful. In bigger animals, where the fingers can be inserted in the cloaca, the sperm extraction can be carried out by pressing around the urogenital papilla with the tip of the fingers or, in smaller animals, with the tip of curved pincers (Figure 9A). One of the advantages of the technique is that it allows to obtain a large amount of sperm in a small amount of time, reducing the level of stress of the animal. However, with this technique it is not possible to obtain the full volume of sperm stored in the seminal vesicle of live animals. Some anatomic features can hamper the use of the abdominal massage, such as the position of the pelvic girdle, or other organs. For example, in *C. monstrosa*, part of the digestive system covers the section of the seminal vesicle where the sperm bundles are more abundant, while in *C. uyato*, the hepatic lobes were large enough to completely cover the seminal vesicles near the pelvic fins, limiting the effectiveness of abdominal massage.

Due to the pressure on the abdominal region, the sperm sample can be spoiled with urine or the contents of the digestive system. Microbial contamination can easily occur if this is the case. Therefore, if the sperm is intended to be fresh stored (medium-term preservation [29]) abdominal massage is not advised. It must be noted that mere passage from the urogenital papilla to the cloaca can also contaminate the samples. Finally, the amount of pressure applied over the reproductive structures should be considered carefully. The entire area surrounding the urogenital papilla is irrigated by capillaries that can be easily damaged, and the internal ducts and structures are fragile. Obviously, when dissection and direct access to the seminal vesicle is possible, all these limitations are easily avoided.

When the amount of sperm in the seminal vesicle is scarce (in immature animals or outside of their reproductive season) or a clean sperm sample is required, cannulation is a better technique to obtain sperm samples. This technique allows the extraction of all the sperm stored in the seminal vesicle. In addition, it allows samples not to be contaminated by microorganisms from the cloaca or the digestive system. However, it is a technique that requires more time than abdominal massage. On the other hand, it requires specialized instruments and can damage the tissue if the anatomy is not well known. Cannulation involves the insertion of a thin tube into the animal body through an orifice, in this case, the urogenital pore (Figure 9B). The diameter of the tube is important as it must be adjusted to the pore size to avoid tissue damage during insertion while keeping a vacuum during the suction. The use of a lubricating jelly with antiseptic can improve the insertion of the tubes while reducing the damage to the tissue and decreasing microbial contamination. The gauge of the cannula should also be selected considering the degree of fluidity or thickness of the sperm. In this study, *H. griseus* and *C. uyato* sperms were extremely dense and the gauge of the cannula had to be increased to get a good sample. In *P. glauca* and *S. canicular*, the sperm fluidity was higher. In *C. monstrosa*, the sperm was so dense (due to the concentration of sperm bundles) in the cranial section of the seminal vesicle that a scarce amount was recovered, and only in the caudal section of the organ was some sperm available.

To make proper cannulation, the position and morphology of the urogenital sinus, seminal vesicle, and ductus deferens must be well known. Otherwise, tissue damage can occur, as the reproductive ducts are very delicate and easy to damage. Perforations in the seminal vesicle can be produced easily, and the catheter can be inserted into the retroperitoneal space and the kidney. If that is the case, blood will be easily seen inside the catheter and the procedure must be stopped immediately. In some species, such as *Cetorhinus maximus* and *Chimaera phantasma* [70,74], the seminal vesicles have transverse folds inside, which can be damaged during the insertion or hamper the extraction procedure. The angle of insertion of the catheter should also be considered and a horizontal angle of 5–10° and a vertical angle of 30° in relation to the axis are advised. For example, in *P. glauca*, there is the possibility of inserting the cannula into the ureter if the cannula deviates. Moreover, in this species, the sperm can be obtained from the urogenital sinus or from the paired ductus deferens if the cannula is inserted deeper.

Once the cannula is inserted, a gentle suction should be enough to make the sperm flow into the syringe. If too much suction is required and there is no advance of sperm through the tube, the cannula could be clogged, or the gauge could be smaller than needed. To prevent the cannula holes from becoming clogged, it is recommended to rotate the cannula gently while the suction is carried out. Moreover, the recommended cannulas are those with lateral perforations and the blunt tip.

The sperm storage in the oviducal gland has been reported in Chondrichthyes (for a brief review, see Marongiu et al. (2015) [75]), being an evolutionarily conserved mechanism to ensure fertilization in nomadic or low population density species [76]. The storage of the sperm occurs in the gland regions called baffle and terminal zone [56], but it has been noted that it is possible to find sperm in other zones, perhaps as a result of recent mating events [75]. A pearly mucus was obtained after scraping of the luminal epithelium of the nidamental glands from *S. canicula*, *S. stellaris*, *P. glauca*, and *C. monstrosa*. The mucus was diluted in an extender solution and aliquots from this dilution were observed under the microscope. The observation revealed the presence of ciliated epithelial cells (moving their cilia), cellular fragments (including spermatozoa remains), and motile sperm in the four species. Although the presence of motile cells could be an indicator of recent mating and not long-term stored sperm, it should be noted that *C. monstrosa* were in their spawning season and in mature phase (stage 3a according to MEDITS category [77]), with no body marks of recent mating (no scrapes or cuts in fins or the body) and with a prolapse in the uteri suggesting recent egg laying. Thus, the mating could hardly be recent. The same can be said for the blue sharks, where motile sperm in the oviducal gland was found in one of the females, only in the left uterus, where also were 25 pups at early stages of development. *P. glauca* males insert just one clasper into the female during mating [59], thus finding only sperm in one of the oviducal glands is plausible. Although the small concentration of motile spermatozoa in the samples could limit the use of this technique as a sperm source to perform artificial insemination, other studies focusing on genetics, spermatozoa morphology, or motility patterns can still be developed.

Chondrichthyan fishes have a higher intrinsic risk of extinction compared to other fish groups [78], which leads to their being one of the most threatened groups on the planet nowadays [10]. Public aquaria play an important role in the conservation of these animals, through ex situ conservation programs [79]. These programs are based on public outreach initiatives and the establishment of an emotional connection between the visitor and the animal [80], by supporting research (including reproductive methods) of different species, and by training professional staff for animal handling and sampling [22]. However, real sustainability of these programs is still far, especially when threatened species are involved [16,81]. Nowadays, public aquaria still rely on captures in the wild or in the spontaneous reproduction of the animals under their care to sustain their zoological collections. Reproductive breeding programs using reproductive techniques, such as sperm extraction and artificial insemination, could allow the advance of public aquaria toward sustainability, but can also be the key to develop reintroduction programs in the wild for threatened species of elasmobranchs. Some shark species have never been able to reproduce in captivity or have done so anecdotally [16]. The captive breeding of one of the most emblematic sharks, the sand tiger shark (*Carcharias taurus*), has only been possible after years of effort and dedication of multiple institutions and researchers, even when this shark species has been kept in aquarium facilities for more than a century [82].

## 4. Conclusions

This work is intended to be a useful guide for veterinarians, aquarists, and researchers who wish to delve into aspects related to the reproduction of these animals. Knowing this specific anatomy in detail is crucial to developing successful protocols for artificial insemination and sperm extraction.

Regarding this technique, although abdominal massage is the simplest technique for obtaining a considerable amount of sperm, it is not the most effective in animals not fully mature or if clean samples are required. Cannulation allows to obtain sperm in a more precise and clean way if the anatomy of the animal is known. Care should be taken during the sample extraction procedure to avoid damage to internal tissues. The dissection of specimens allows samples to be obtained directly from the sperm storage structures, such as seminal vesicles, sperm sacs, and deferent ducts. Finally, obtaining active sperm from the oviducal gland in females opens new research opportunities that should be exploited in the future. Much work remains to be done on the development and application of reproductive techniques in Chondrichthyes, but these first steps can be crucial for the future conservation of these animals.

## Figures and Tables

**Figure 1 animals-11-02191-f001:**
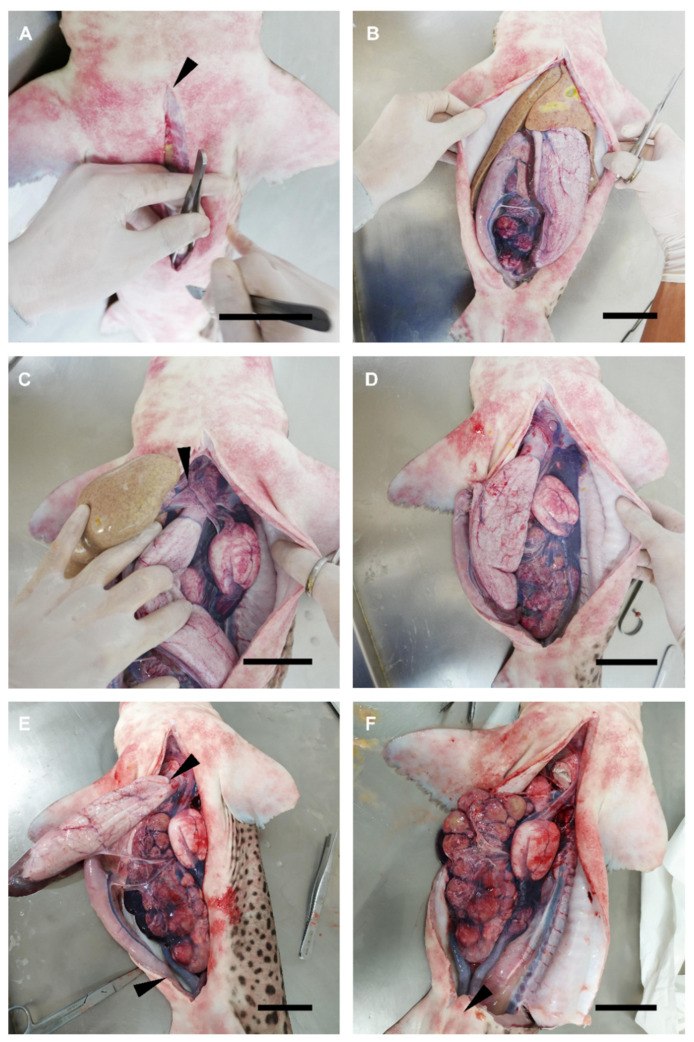
Dissection procedure. Relevant steps in the dissection procedure of a female nursehound (*Scyliorhinus stellaris*) to reach the reproductive system. (**A**) Longitudinal incision over the pectoral girdle of the coracoid bar following the ventral midline. Notice the use of forceps to elevate the abdominal wall. (**B**) General vision of the pleuroperitoneal cavity. (**C**) Lateral displacement of the liver, esophagus, and stomach to reach the falciform ligament. (**D**) General vision of the digestive and reproductive systems. (**E**) Lateral displacement of the esophagus, stomach, spiral valve, and rectum and cut on the esophagus and rectum. (**F**) General vision of the reproductive system. Arrowheads mark incision areas: (**A**) coracoid bar, (**C**) falciform ligament, and (**E**) esophagus (superior) and rectum (inferior); (**F**) puboischiadic bar. The scale bar indicates 6 cm.

**Figure 2 animals-11-02191-f002:**
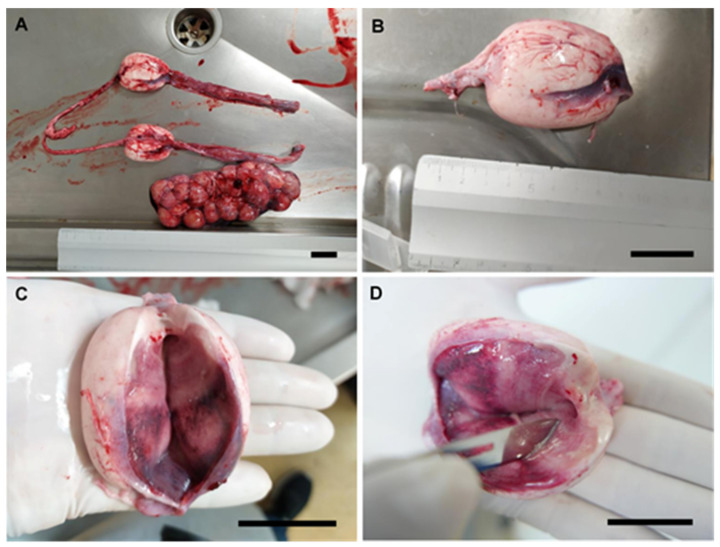
Sperm recovery from nidamental gland. Nidamental glands from a nursehound, *Scyliorhinus stellaris* (**A**), united by the oviducts and ostium. The gland was removed and cleaned (**B**) and split through its cranial–caudal axis, exposing its lumen (**C**). Scraping was done over its luminal epithelium using the edge of a scalpel (**D**) to collect a pearly mucus containing sperm. The scale bar indicates 3 cm.

**Figure 3 animals-11-02191-f003:**
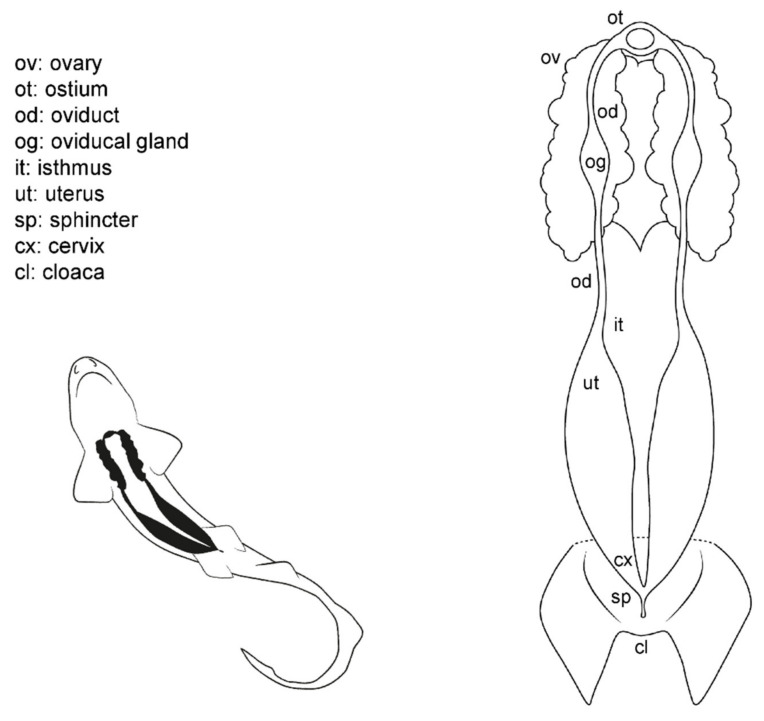
Female general anatomy: sharks. Morphological scheme of an ideal female shark, showing the main reproductive structures: ovary (ov), oviducts (od), ostium (ot), oviducal glands (og), isthmus (it), uterus (ut), sphincter (sp), cervix (cx), and cloaca (cl).

**Figure 4 animals-11-02191-f004:**
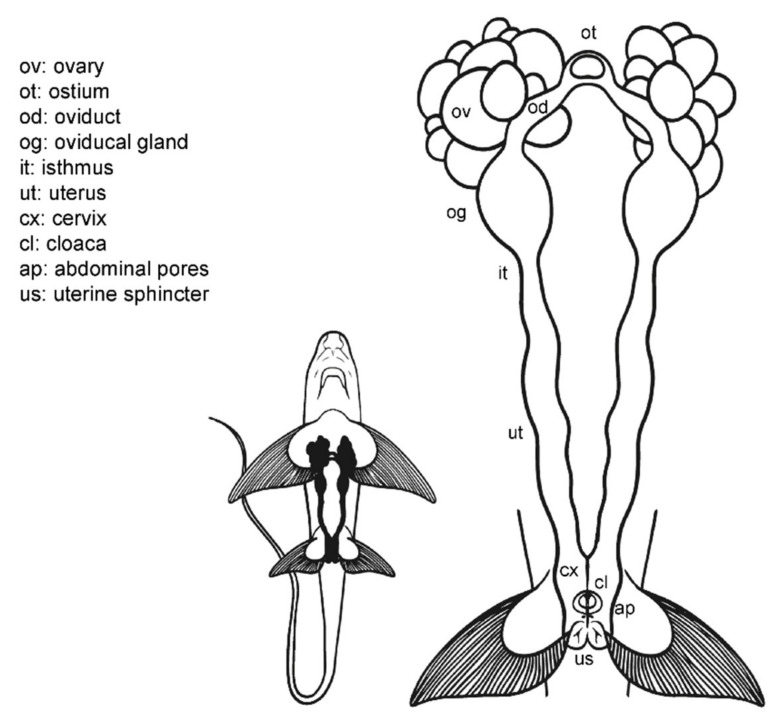
Female general anatomy: *Chimaera monstrosa*. Morphological scheme of *C. monstrosa* main female reproductive structures: ovary (ov), ostium (ot), oviduct (od), oviducal gland (og), isthmus (it), uterus (ut), cervix (cs), cloaca (cl), abdominal pores (ap), and uterine sphincter (us).

**Figure 5 animals-11-02191-f005:**
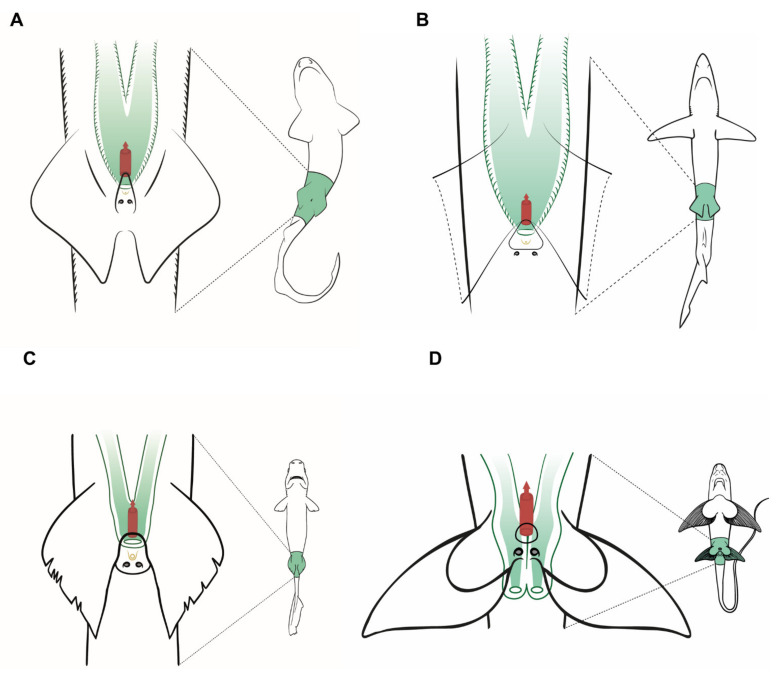
Specific female anatomy. Species-specific morphologies in three female sharks and one chimaera species that can be relevant to perform artificial insemination. The reproductive, excretory, and digestive systems are marked with different colors. The uteri are shown in green, the excretory system in yellow, and the access to the digestive system in red. The grey circles are the abdominal pores that connect the pleuroperitoneal cavity with the exterior. (**A**) Model for catsharks, family Scyliorhinidae. (**B**) Blue shark *Prionace glauca*. (**C**) Model observed in velvet belly lanternshark *Etmopterus spinax*. (**D**) Rabbitfish *Chimaera monstrosa*.

**Figure 6 animals-11-02191-f006:**
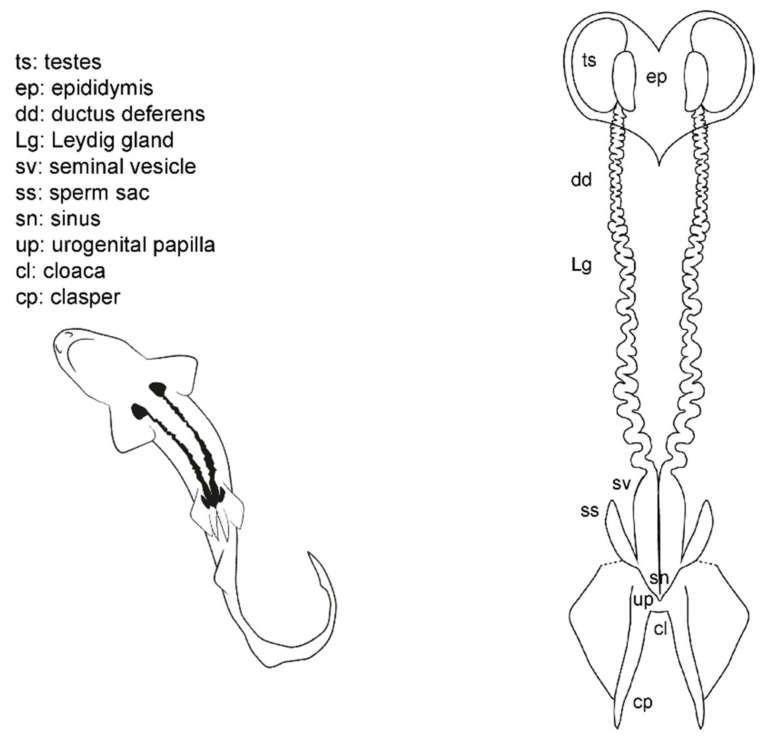
Male general anatomy: sharks. Morphological scheme of an ideal shark male, showing the main reproductive structures: testes (ts), epididymis (ep), ductus deferens (dd), Leydig gland (Lg), seminal vesicle (sv), sperm sac (ss), sinus (sn), urogenital papilla (up), cloaca (cl), and clasper (cp).

**Figure 7 animals-11-02191-f007:**
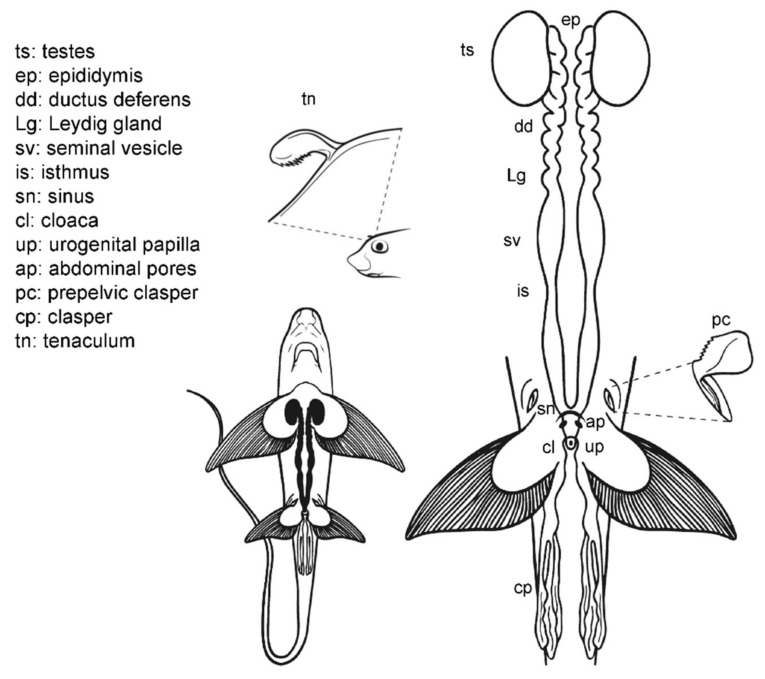
Male general anatomy: *Chimaera monstrosa*. Morphological scheme of the main reproductive male structures of *C. monstrosa*: testes (ts), epididymis (ep), ductus deferens (dd), Leydig gland (Lg), seminal vesicle (sv), isthmus (is), sinus (sn), cloaca (cl), urogenital papilla (up), abdominal pores (ap) and claspers [cephalic or tenaculum (tn), prepelvic (pc) and pelvic (cl)].

**Figure 8 animals-11-02191-f008:**
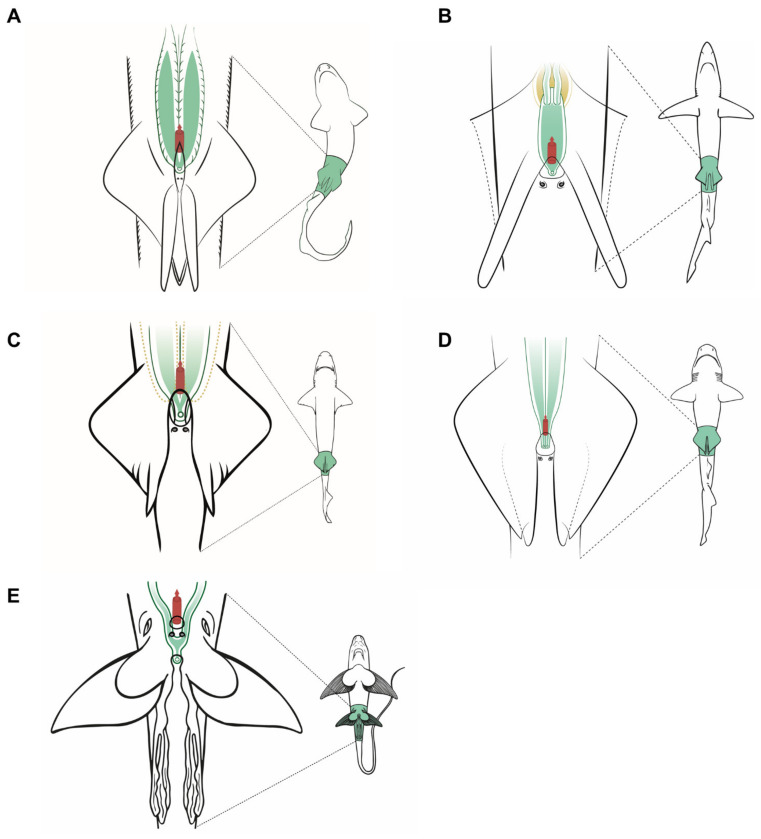
Specific male anatomy. Species-specific morphologies in sharks and *C. monstrosa* with relevance in sperm extraction procedures. The reproductive, excretory, and digestive systems are marked with different colors: the seminal vesicle and ductus deferens in green, the excretory system in yellow, and the access to the digestive system in red. The dotted lines in *Centrophorus uyato* represent the position of the hepatic lobes. The grey circles are the abdominal pores that connect the pleuroperitoneal cavity with the exterior. (**A**) Model observed in catsharks, family Scyliorhinidae. (**B**) Model for blue shark *Prionace glauca*. (**C**) Model observed in little gulper shark *Centrophorus uyato*. (**D**) Bluntnose sixgill shark *Hexanchus griseus*. (**E**) Rabbitfish *Chimaera monstrosa*.

**Figure 9 animals-11-02191-f009:**
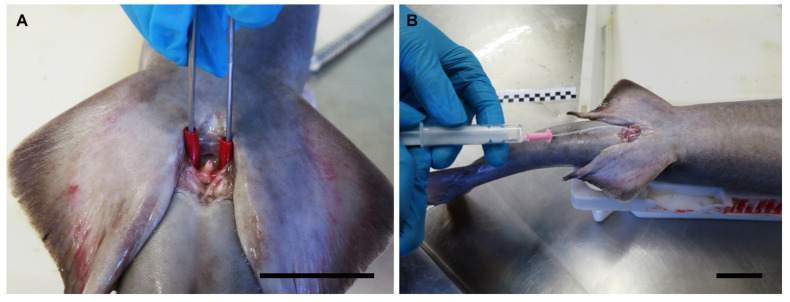
Different sperm extraction procedures in the little gulper shark (*Centrophorus uyato*). (**A**) Massage over the urogenital sinus using forceps with rubber tips in species where abdominal massage is inefficient—in this case, because of the hepatic lobes covering the seminal vesicles. (**B**) Cannulation through the urogenital papilla using a catheter. The scale bar indicates 3 cm.

**Table 1 animals-11-02191-t001:** Species in the study. Number of males (NM) and females (NF) from each species, size range of the specimens used, and their origin and conservation status according to IUCN (International Union for Conservation of Nature) criteria for the Mediterranean: not evaluated (NE), least concern (LC), near threatened (NT), vulnerable (VU), endangered (EN), and critically endangered (CR). Animals in aquaria (AQ) were part of the Oceanogràfic zoological collection. Animals from commercial fisheries were captured by gill net or bottom trawling and sold in fish markets (FM) or discarded as by-catch (BC). Stranded animals (ST) were recovered by the Comunitat Valenciana Stranding Network.

Common Name	Scientific Name	NM	NF	IUCN	Source	Range (cm)
Small-spotted catshark	*Scyliorhinus canicula*	7	5	LC	AQ/FM/BC	38–57
Nursehound	*Scyliorhinus stellaris*	7	3	NT	AQ	75–144
Blackmouth catshark	*Galeus melastomus*	4	6	LC	BC/FM	48–69
Blue shark	*Prionace glauca*	2	1	CR	ST	290–297
Velvet belly lanternshark	*Etmopterus spinax*	-	4	LC	BC	33–40
Little gulper shark	*Centrophorus uyato*	1	-	NE	BC	86
Bluntnose sixgill shark	*Hexanchus griseus*	1	-	LC	ST	250
Rabbitfish	*Chimaera monstrosa*	2	2	NT	BC	104/112

## Data Availability

Not applicable.

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
