# Peer review of "Reproductive Anatomy of Chondrichthyans: Notes on Specimen Handling and Sperm Extraction. II. Sharks and Chimaeras"

_animals, 2021, doi:10.3390/ani11082191_

Round 1

Reviewer 1 Report

The manuscript has been revised according to the reviewer's comments and can be published. 

Author Response

Thanks a lot for your comments

Reviewer 2 Report

My comments are the same as on the twin paper on rays and rays.

The main content is the anatomy of the genitals, especially the male genitals of a number of shark and ratfish. This is, however, not something that has not been studied before. The current finding need to be compared with earlier studies (even if they are old and in other languages) to see what is new, confirmatory or in contrast to the previous studies. This is general demand for scientific articles.

The practical importance of the findings is only of interest for animals that can be readily kept in captivity anyway. Also, the success of the sperm sampling is not presented consistently and the quality of the sperm in the samples is dealt with very scarcely.

Author Response

The main content is the anatomy of the genitals, especially the male genitals of a number of shark and ratfish. This is, however, not something that has not been studied before. The current finding need to be compared with earlier studies (even if they are old and in other languages) to see what is new, confirmatory or in contrast to the previous studies. This is general demand for scientific articles.

As you suggested, we have included relevant references and compared them to our findings. There are plenty studies about the reproduction of these group of animals. However, some of the literature (including those studies on morphology) are focused on histology, organ development, or other aspects not useful to our study and therefore were not included.

The practical importance of the findings is only of interest for animals that can be readily kept in captivity anyway. Also, the success of the sperm sampling is not presented consistently and the quality of the sperm in the samples is dealt with very scarcely.

The exposed techniques not only are useful for those animals that can be readily kept in captivity. Our research group is working with cell cryopreservation protocols, which require sperm in optimal conditions. But there are more studies where sperm extraction protocols should be considered: i) physiology of spermatozoa and seminal plasma; ii) pattern and dynamics of movement and sperm displacement; iii) taxonomic value of spermatozoa morphology; iv) impact of environment (temperature, pH, drugs) on sperm quality over time, etc. However, even if this was the case, the diversity of species kept in zoological collections is not negligible and the professionals working with them can used the techniques presented in the manuscript.

Part of the text describing the sperm extraction processes has been rewritten to clarify the information.

Reviewer 3 Report

Dear Authors,

I personally had already appreciated your manuscript at the first submission, I note with pleasure that you have also taken into account my previous minor revisions, seriously following the suggestions given.

For this reason in my opinion the manuscript is ready for publication.

Good job!

Best regards

The reviewer

Author Response

Thanks a lot for your comments.

This manuscript is a resubmission of an earlier submission. The following is a list of the peer review reports and author responses from that submission.

Round 1

Reviewer 1 Report

OVERALL COMMENTS:

The manuscript of Pablo García-Salinas, Victor Gallego and Juan F. Asturiano focusing on a very interesting topic with good accuracy and quality. The reading is clearly and sliding capturing the reader into the topic; the methodologies utilized in the study are described simply and enriched with images and figures that show in detail the technique that can be used by other researcher to survey and purse the same aim of the present study. Basing on the overexploitation of the Chondrichthyes, this study should be considered a novelty for the conservation of the species. As already argued by Authors in the conclusion section, this study should be considered an anatomical guide as one of the main required tools for the conservation of Chondrichthyes (related to Sharks and Chimaeras).  

In my opinion, due to the novelty, writing and drafting, background, methodologies and accuracy applied by the Authors of this study, this paper could be accept for publication after some minor revisions.

All the specific comments about the paper are listed in the sections below.

Introduction:

The introduction section present a good background, however, to increase in interest to readers and consistency to the aim of the study, I suggest to Authors to better introduce and increase in knowledge about the exploitation, overfishing and finning events of the Chondrichthyes specially for elasmobranchs, through the use of further references to support, such as: 

D’Iglio C., Savoca S., Rinelli P., Spanò N., Capillo G. Diet of the Deep-Sea Shark Galeus melastomus Rafinesque, 1810, in the Mediterranean Sea: What We Know and What We Should Know. Sustainability, 2021 13(7), 3962.

Dulvy, N.K., Allen, D.J.; Ralph, G.M.; Walls, R.H.L. The Conservation Status of Sharks, Rays and Chimaeras in the Mediterranean Sea; IUCN: Malaga, Spain, 2016.

Moreover, the section could be improved adding further references on the use of some Elasmobranch as model for anatomy studies, such as: 

Coolen, M., Menuet, A., Chassoux, D., Compagnucci, C., Henry, S., Lévèque, L., … Wincker, P. (2008). The dogfish Scyliorhinus canicula: A reference in jawed vertebrates. Cold Spring Harbor Protocols, 2008(12), pdb. emo111.

Lauriano E. R., Pergolizzi S., Gangemi J., Kuciel M., Capillo G., Aragona M., Faggio C. Immunohistochemical colocalization of G protein alpha subunits and 5‐HT in the rectal gland of the cartilaginous fish Scyliorhinus canicula. Microscopy Research and Technique, 2017

Line-43, 353, 354, 355, 358, 422: For a better form of the manuscript presentation, I suggest to Authors to leave the citation outside the round brackets.

Materials and Methods:

In this section, the methodologies are well explained in detail, is very interesting for the reader discover the techniques and the methods, sometimes “home-made”, used to sampling the specimens.

In the Table 1 the species are clearly categorized, however, in the caption the meaning of “ST” for the two species Prionace glauca and Hexanchus griseus were missed.

I think the mistake came from by the web system but in Figure 1 only A and B are visible, the figures from C to F were missed. Furthermore, I suggest to Authors to shift the position of Figure 1 close to the section 2.2.

Line 167-176: I suggest to Authors to specify, possibly with an additional Table, the number of specimens and relative sperm extraction method utilized. 

Results and Discussion:

the section is clearly and complete, data are well showed.

Line 537-541: The handling specimens discussed in this section came from both “AQ, FM, BC, ST”, could the different origins influence the sperm density? Referring to their showed experience (line537), did the Authors mean in this study or along theirs career? In this case I suggest to Authors to increase in literature about this topic.

Best Regards

The Reviewer

Author Response

Reviewer 1:

The introduction section present a good background, however, to increase in interest to readers and consistency to the aim of the study, I suggest to Authors to better introduce and increase in knowledge about the exploitation, overfishing and finning events of the Chondrichthyes specially for elasmobranchs, through the use of further references to support, such as:

D’Iglio C., Savoca S., Rinelli P., Spanò N., Capillo G. Diet of the Deep-Sea Shark Galeus melastomus Rafinesque, 1810, in the Mediterranean Sea: What We Know and What We Should Know. Sustainability, 2021 13(7), 3962.

Dulvy, N.K., Allen, D.J.; Ralph, G.M.; Walls, R.H.L. The Conservation Status of Sharks, Rays and Chimaeras in the Mediterranean Sea; IUCN: Malaga, Spain, 2016.

Thank you for your suggestion. We have included a paragraph focusing on the conservation status of sharks, rays and chimeras, adding new references and highlighting the importance of the Mediterranean as an extinction hotspot.

“Regarding to their conservation status, Chondrichthyes possess life histories that make them sensitive to elevated anthropic pressure, threatening their populations [7,8]. In fact, chondrichthyan extinction risk is higher than for most other vertebrates, and only one-third of the species assessed are considered safe, according to IUCN Red list criteria [9] The situation is particularly sensitive in places like the Mediterranean Sea, a key hotspot of extinction risk, where half the species of rays and sharks face an elevated risk of extinction [10]. Among the drivers for the global decline of its populations, overfishing (intentional or incidental) and habitat destruction are the main causes [9-11].”

Moreover, the section could be improved adding further references on the use of some Elasmobranch as model for anatomy studies, such as:

Coolen, M., Menuet, A., Chassoux, D., Compagnucci, C., Henry, S., Lévèque, L., … Wincker, P. (2008). The dogfish Scyliorhinus canicula: A reference in jawed vertebrates. Cold Spring Harbor Protocols, 2008(12), pdb. emo111.

Lauriano E. R., Pergolizzi S., Gangemi J., Kuciel M., Capillo G., Aragona M., Faggio C. Immunohistochemical colocalization of G protein alpha subunits and 5‐HT in the rectal gland of the cartilaginous fish Scyliorhinus canicula. Microscopy Research and Technique, 2017

Thank you very much for your suggestion. We have modified the introduction by adding information and references about the use of elasmobranchs as animal models for studies on morphology and physiology.

“Due their position as one of the oldest vertebrated groups, elasmobranchs have been previous used as animal models for physiological and morphological studies [29-32]. However, some details about the morphology of certain reproductive structures, important during the use of reproductive techniques, have not been previously considered. Thus, the main objective of this study is to offer a useful guide of reproductive system anatomy of sharks and chimaeras, intending to be useful during sperm obtention procedures, and propose preliminary indications on the female anatomy to be considered during artificial insemination practices.”

Line-43, 353, 354, 355, 358, 422: For a better form of the manuscript presentation, I suggest to Authors to leave the citation outside the round brackets.

As you suggest, for a better reading, we have left the citation outside the round brackets:

“ The absence of cloaca is described for most Holocephali [47,48], but some species such the elephant fish (Callorhynchus milli) [49], the cockfish (Callorhynchus callorhynchus)[50], or the Eastern Pacific black ghostshark (Hydrolagus melanophasma) [51] possess a common cloaca where the anus and the uterine openings are found. Also, a sperm pouch for sperm storage can be found on these species, but it is absent in C. monstrosa. Another reproductive trait, the prepelvic clasper pouch (present in female Callorhynchus species) [48–50] is not present in C. monstrosa females.

Materials and Methods:

In this section, the methodologies are well explained in detail, is very interesting for the reader discover the techniques and the methods, sometimes “home-made”, used to sampling the specimens.

In the Table 1 the species are clearly categorized, however, in the caption the meaning of “ST” for the two species Prionace glauca and Hexanchus griseus were missed.

The missing of the abbreviature was a mistake, we have added the following line:

Stranded animals (ST) were recovered by the Comunitat Valenciana Stranding Network.

Thanks for noticing.

I think the mistake came from by the web system but in Figure 1 only A and B are visible, the figures from C to F were missed. Furthermore, I suggest to Authors to shift the position of Figure 1 close to the section 2.2.

Indeed, there have been some unintentional movement of the Figure 1 inside the manuscript. We have moved the figure to a position closer to section 2.2. Thanks for the suggestion.

Line 167-176: I suggest to Authors to specify, possibly with an additional Table, the number of specimens and relative sperm extraction method utilized.

Table 1 already shows the number of specimens used for the sperm extraction procedures, so instead of using a new table, we have modified the following paragraph to make clear the fact that the three sperm extraction methods were used in every species used:

“Three different methods were used to obtain sperm form dead males in every species: i) abdominal massage on the ventral region immediately anterior to the pelvic girdle, or by pressing around the urogenital papilla in the cloacal cavity with the fingers or with curved pincers (only in sharks), ii) accessing by dissection and stripping directly on the seminal vesicle. In both cases sperm flowing from the urogenital papilla was immediately collected using a sterile syringe, plastic tube or a pipette. And iii) through cannulation, by introducing a polyurethane cat catheter….”

Results and Discussion:

the section is clearly and complete, data are well showed.

Line 537-541: The handling specimens discussed in this section came from both “AQ, FM, BC, ST”, could the different origins influence the sperm density?

This is an interesting question. There are some species for which we cannot know how their different origins affect sperm, since we only have one specimens source. However, for the shark Scyliorhinus canicula and in certain species of rays of the genus Raja, we have animals of both aquarium, fish markets, and by-catch.

There does not appear to be any difference, at least in the density value, that we can attribute to the origin of the sample.  And the differences are probably due to species differences.

Referring to their showed experience (line537), did the Authors mean in this study or along theirs career? In this case I suggest to Authors to increase in literature about this topic.

The comments about sperm density referred only to this study. To clarify it, we have changed the previous sentence to:

“In this study, H. griseus and C. uyato sperm was extremely dense and the gauge of the cannula had to be increased to get a good sample.”

Unfortunately, few research has been done studying in detail the characteristics of the sperm in these diverse group of animals. Information about sperm composition, density, viscosity and spermatozoa morphology is still scarce.

Reviewer 2 Report

Review of  “Reproductive anatomy of Chondrichthyans: notes on specimen handling and sperm extraction. II. Sharks and chimaeras.

The general aim of the study is to develop techniques for artificial fertilization of rays and skates for conservational purposes.

A major part of the article consists of gross anatomical descriptions of the reproductive organs of several species of rays and skates.  

There are probably masses of literature on the general anatomy of the reproductive organs of sharks and chimaeras. Much of it is probably in old articles and in several languages. Nevertheless, the findings in the present article must carefully compared with what is already known, which is far from the case now. What agrees with earlier descriptions, what does not? The discussion ought to focus on what is new.

The article describes how sperm were sampled from living or more or less freshly dead skates and rays, most of them discarded as bycatch or bought at fish markets. However, as a technical note this is of no use unless there is information on how well the artificial fertilizations went. There is no such information in the MS. I guess that the authors have this information (if they don’t, all is worthless) and intend to publish that in a separate paper, but I do not think that such a dividing up is justified.

Other points

The whole idea of using artificial insemination is of course only useful in species that could be kept in captivity. How is that with the different species used?     

The English is sometimes strange. e.g.:

Line 110. Shelled? Sold?

I recommend that someone with a really good knowledge of English, preferably a native speaker, should go over the text.

The conclusions are mainly about things already known before the study, it ought to be conclusions from the present study.

Author Response

Reviewer 2:

The general aim of the study is to develop techniques for artificial fertilization of rays and skates for conservational purposes.

A major part of the article consists of gross anatomical descriptions of the reproductive organs of several species of rays and skates. 

There are probably masses of literature on the general anatomy of the reproductive organs of sharks and chimaeras. Much of it is probably in old articles and in several languages. Nevertheless, the findings in the present article must carefully compared with what is already known, which is far from the case now. What agrees with earlier descriptions, what does not? The discussion ought to focus on what is new.

As you said, it is true that there are old articles which describe the anatomy of these animals. But the objective of this work is on detailing those anatomic structures that will have interest when practicing specific procedures. On the other hand, the old literature that exists is sometimes difficult to obtain and as you said, can be found in several languages (mainly in German). In addition, practical issues (such as the number and position of the urogenital pores) are often excluded from this type of studies.

But we should insist (and it is specified in the text), the objective of our study is very specific. The article is intended to be useful to aquarists, veterinarians, or other researchers who wish to extract sperm in an efficient and accurate manner, without affecting the animals more than necessary.

The article describes how sperm were sampled from living or more or less freshly dead skates and rays, most of them discarded as bycatch or bought at fish markets. However, as a technical note this is of no use unless there is information on how well the artificial fertilizations went. There is no such information in the MS. I guess that the authors have this information (if they don’t, all is worthless) and intend to publish that in a separate paper, but I do not think that such a dividing up is justified.

The possibility of extracting viable gametes, in this case sperm, allows the achievement of other objectives, beyond immediate artificial insemination.

For example, our research group is working with cells cryopreservation protocols, which require sperm in optimal conditions. The success of cryopreservation is affected by sperm quality, so the usefulness of proper sperm extraction protocols cannot be neglected. But there are more studies where sperm extraction protocols should be considered: i) physiology of spermatozoa and characteristics of seminal plasma; ii) pattern and dynamics of movement and spermatozoa displacement; iii) taxonomic value of spermatozoa morphology; iv) impact of environmental parameters (temperature, pH, drugs) on sperm quality over time, etc. Therefore, while it is true that artificial insemination requires good sperm extraction procedures, other projects benefit from correct handling during the extraction of gametes.

The tools we are designing in our research group (such as sperm extraction in males, sperm extraction in females, or sperm cryopreservation) may never will be used in endangered species. However, if the moment arrive, it would be great being prepared.

Other points

The whole idea of using artificial insemination is of course only useful in species that could be kept in captivity. How is that with the different species used?    

Although it is true that the insemination of species can be useful in animals kept in captivity, it is not the only use of this technique. Insemination of wild females may be completely feasible, especially in populations of sparsely dispersed species or those with a population bias (by number of males or their efficacy) in sexes. In fact, population control, including artificial insemination, can be a valid management tool in certain situations.

On the other hand, the diversity of species kept in zoological collections is not negligible. While it is true, to our knowledge, that not all the species with which we have worked in this study are part of the zoological collections of research centres or aquariums, the knowledge obtained from them is equally valid and can be extrapolated to other species that are in these centres. As an example, we do not know of any aquarium that exhibits Centrophorus in its collections. However, other close relatives, the squaliforms Oxynotus or Squalus, are kept in captivity. The same happens in the case of Hexanchus, which although is only occasionally exhibited by some aquaria, has a close relative, Notorynchus, whose reproduction is long sought in captivity.

The English is sometimes strange. e.g.:

Line 110. Shelled? Sold?

These errors and other grammatical inaccuracies have been corrected in the text.

 I recommend that someone with a really good knowledge of English, preferably a native speaker, should go over the text.

The conclusions are mainly about things already known before the study, it ought to be conclusions from the present study.

We have modified the section according to your suggestion.

Reviewer 3 Report

The manuscript provided the detail of reproductive anatomy of chondrichthyans, which is quite clear to see structure and the sperm extraction. The data is solid and useful for the ex situ conservation by artificial insemination. The English writing style and format is acceptable for publication. Therefore, it is qualified for the Journal of Animals.

Scale bar should provide to show the size of sample in the figure 1, 2, 9.

Author Response

Reviewer 3:

The manuscript provided the detail of reproductive anatomy of chondrichthyans, which is quite clear to see structure and the sperm extraction. The data is solid and useful for the ex situ conservation by artificial insemination. The English writing style and format is acceptable for publication. Therefore, it is qualified for the Journal of Animals.

Scale bar should provide to show the size of sample in the figure 1, 2, 9.

We have provided a scale bar for the figures as requested. Thanks for the suggestion.

Reviewer 4 Report

This manuscript is another excellent piece of applied comparative anatomy, It can be published after very minor corrections.

“In the case of medium size and small sharks, the animals were flipped dorsally exposing its their ventral surface”.

“A small incision was made over the coracoid bar of the pectoral girdle and a longitudinal cut was made along the medial line towards the pelvic girdle, over the cloaca (Figure 1A)”.  I think the correct word is median not medial.

I would suggest to use the terms cranial and caudal instead of anterior and posterior throughout the text.

Fig. 1.

“Necropsy procedure. Relevant steps in the necropsy procedure of a female nursehound (Scyliorhinus stellaris) to reach the reproductive system. A) longitudinal incision over the pectoral girdle of the coracoid bar following the medial line”. Again, I think the correct word is median not medial.

“lateral displacement of the liver and digestive system to reach the anterior mesenteries”. The digestive system includes organs not affected by the necroscopy. I would specify which organs were displaced.  Why mesenteries instead of mesentery?

“… the terminal section of the uteri (cervix) can converge and be fused in a common cervix (or urogenital sinus) or be independent. In both situations, the access to the uteri is closed by a sphincter. The last orifice inside the cloaca is the distal section of the digestive system: the rectum and the anus”.  I would change the sentence as follows: the caudal end of the uteri can be independent or converge and fuse in a common cervix (or urogenital sinus).  The caudal orifice inside the cloaca …..………..

“At the base of the pectoral fins two independent orifices closed by a sphincter are the entrances to the uteri. In juvenile females the orifices are narrow and hard to find, while in mature females are easier two to locate.”

“The catheter with the sperm sample should pass through different anatomical barriers (such as cloacal morphology, cervix, or uterine sphincters) that can hamper the overall insemination procedure [20]”.  I would suggest to re-phrase this sentence because morphology is not itself a barrier.

“The cloaca is located between the pelvic fins and it is covered by its inner margins…..” It is not clear what “..covered by its inner margins” means.

“Near the cephalic region two paired testes embedded in the epigonal organ are connected via efferent ductules….”. What does “near the cephalic region” means? Is the cephalic region defined in the manuscript or in the literature? Maybe better “The paired testes are located at the cranial end of the reproductive trait…” or something like that.

“The gland is formed by the anterior part of the mesonephros and has a secretory function, producinges the seminal fluid and the matrix where the spermatozeugmata”. Glands have secretory functions by definition, so the sentence can be reduced.

“The siphon sacs in the three species are also visible under the ventral skin above the pelvic fins”. The sentence is not clear: 1) the pelvic fins are located in the ventral side of the body; 2) above means dorsal not cranial (anterior);  so maybe “The siphon sacs in the three species are also visible under the skin cranial to the pelvic fins” .  

“The epigonal organ, present in elasmobranchs, is absent in C. mon-486 strosa and probably in the rest of holocephalans [45,51,54].” Any explanation for this absence?

“The observation revealed the presence of ciliated epithelial cells (moving its their cilia)…”.

Author Response

Reviewer 4:

“In the case of medium size and small sharks, the animals were flipped dorsally exposing its their ventral surface”.

The word has been changed as you suggested.

“A small incision was made over the coracoid bar of the pectoral girdle and a longitudinal cut was made along the medial line towards the pelvic girdle, over the cloaca (Figure 1A)”.  I think the correct word is median not medial.

To avoid confusion, it has been changed for “ventral midline” through the entire text.

I would suggest to use the terms cranial and caudal instead of anterior and posterior throughout the text.

It has been changed as you suggest thorough the entire text. The only exception has been made when describing oviducts, because the terms “anterior/ posterior oviduct” has been used by previous authors.

Fig. 1.

“Necropsy procedure. Relevant steps in the necropsy procedure of a female nursehound (Scyliorhinus stellaris) to reach the reproductive system. A) longitudinal incision over the pectoral girdle of the coracoid bar following the medial line”. Again, I think the correct word is median not medial.

To avoid confusion, it has been changed for “ventral midline” through the entire text.

“lateral displacement of the liver and digestive system to reach the anterior mesenteries”. The digestive system includes organs not affected by the necroscopy. I would specify which organs were displaced.  Why mesenteries instead of mesentery?

The text has been modified as:

“lateral displacement of the liver, esophagus and stomach to reach the falciform ligament”

“… the terminal section of the uteri (cervix) can converge and be fused in a common cervix (or urogenital sinus) or be independent. In both situations, the access to the uteri is closed by a sphincter. The last orifice inside the cloaca is the distal section of the digestive system: the rectum and the anus”.  I would change the sentence as follows: the caudal end of the uteri can be independent or converge and fuse in a common cervix (or urogenital sinus).  The caudal orifice inside the cloaca …..………..

The text has been changed as you suggested:

“Inside the cloaca, the caudal end of the uteri can be independent or converge and fuse in a common cervix (or urogenital sinus).  The cranial orifice inside the cloaca is the caudal section of the digestive system: the rectum and the anus.”

“At the base of the pectoral fins two independent orifices closed by a sphincter are the entrances to the uteri. In juvenile females the orifices are narrow and hard to find, while in mature females are easier two to locate.”

The text has been modified as:

“The entrances to the uteri are two independent orifices closed by a sphincter located at the base of the pectoral fins. In juvenile females the orifices are narrow and hard to find, while in mature females are easier to locate.”

“The catheter with the sperm sample should pass through different anatomical barriers (such as cloacal morphology, cervix, or uterine sphincters) that can hamper the overall insemination procedure [20]”.  I would suggest to re-phrase this sentence because morphology is not itself a barrier.

The text has been modified following your suggestion:

“To be able to deposit a sperm sample in the correct oviduct, the catheter needs to be inserted through different structures (such as cloaca, uterine sphincters or cervix) that can hamper the overall process.”

“The cloaca is located between the pelvic fins and it is covered by its inner margins…..” It is not clear what “..covered by its inner margins” means.

The text has been modified following to make the sentence clearer:

“The cloaca is located between the pelvic fins. Typically, is covered by the inner margins of the pelvic fins (the metapterygia) or in some species by two cloacal lips.”

“Near the cephalic region two paired testes embedded in the epigonal organ are connected via efferent ductules….”. What does “near the cephalic region” means? Is the cephalic region defined in the manuscript or in the literature? Maybe better “The paired testes are located at the cranial end of the reproductive trait…” or something like that.

The text has been modified following your suggestion:

“The paired testes, embedded in the epigonal organ, are located at the cranial end of the reproductive trait. The testes are connected….”

“The gland is formed by the anterior part of the mesonephros and has a secretory function, producinges the seminal fluid and the matrix where the spermatozeugmata”. Glands have secretory functions by definition, so the sentence can be reduced.

It has been modified following your suggestion:

“The gland is formed by the cranial part of the mesonephros and produces the seminal fluid and the matrix where the spermatozeugmata, or the spermatophores, are formed.”

“The siphon sacs in the three species are also visible under the ventral skin above the pelvic fins”. The sentence is not clear: 1) the pelvic fins are located in the ventral side of the body; 2) above means dorsal not cranial (anterior);  so maybe “The siphon sacs in the three species are also visible under the skin cranial to the pelvic fins” . 

The text has been modified following your suggestion:

“The siphon sacs in the three species are also visible under the skin cranial to the pelvic fins”

“The epigonal organ, present in elasmobranchs, is absent in C. mon-486 strosa and probably in the rest of holocephalans [45,51,54].” Any explanation for this absence?

The epigonal organ in elasmobranchs is made up of lymphoid and hemopoietic tissue. In chimeras the greatest concentration of this type of tissue occurs in certain structures of the head, unique in these animals. It could be that the absence of the epigonal organ is due to the fact that its function is already performed in this specific organ. You can find more information about it in:

Mattisson, A., Fänge, R., and Zapata, A. (1990). Histology and ultrastructure of the cranial lymphohaemopoietic tissue in Chimaera monstrosa (Pisces, Holocephali). Acta Zool. 71, 97–106.

“The observation revealed the presence of ciliated epithelial cells (moving its their cilia)…”.

It has been modified following your suggestion.